# Evolution of the Management of Brain Metastases: A Bibliometric Analysis

**DOI:** 10.3390/cancers15235570

**Published:** 2023-11-24

**Authors:** Ikram A. Burney, Aya H. Aal Hamad, Syed F. A. Hashmi, Nisar Ahmad, Nadeem Pervez

**Affiliations:** 1Sultan Qaboos Comprehensive Cancer Care and Research Center, Muscat 123, Oman; i.burney@cccrc.gov.om (I.A.B.); s.hashmi@cccrc.gov.om (S.F.A.H.); 2Panjwani Center for Molecular Medicine & Drug Research, International Center of Chemical and Biological Sciences, Karachi University, Karachi 75270, Pakistan; 3Sultan Qaboos University Hospital, Muscat 112, Oman; ayaalhamad96@gmail.com; 4Department of Economics and Finance, College of Economics and Political Science, Sultan Qaboos University, Muscat 123, Oman; nisar@squ.edu.om

**Keywords:** brain metastasis, CNS metastasis, randomized controlled trial, integrative review, SRS, Whole Brain Radiotherapy (WBRT), hippocampal sparing radiotherapy, Simultaneous Integrated Boost (SIB), Oman

## Abstract

**Simple Summary:**

The brain is a common site of metastases from cancer. Approximately 10% of all patients develop brain metastasis during their illness. Brain metastasis used to be frequently associated with significant morbidity and short survival, measured in months. However, with advances in treatment, a significant number of patients can now expect to live for several years, with a better quality of life. We reviewed the literature published over the last 50 years, to identify patterns of care of patients diagnosed with brain metastases. Techniques such as whole brain radiotherapy are used sparingly only. A more in-depth knowledge about the cancers, and advances in radiotherapy techniques, such as focused radiation to the site of metastases, and targeted therapy which crosses the barrier between the blood and the brain have revolutionized the care, especially in patients with lung cancer, breast cancer, and melanoma. This type of literature review not only helps to summarize the evolution of clinical practice, but also helps to identify the current trends for researchers.

**Abstract:**

A systematic review of the published literature was conducted to analyze the management evolution of brain metastases from different cancers. Using the keywords “brain metastasis”, “brain metastases”, “CNS metastasis”, “CNS metastases”, “phase III” AND/OR “Randomized Controlled Trial” (RCT), relevant articles were searched for on the SCOPUS database. A total of 1986 articles were retrieved, published over a 45-year period (1977–2022). Relevant articles were defined as clinical studies describing the treatment or prevention of brain metastases from any cancer. Articles on imaging, quality of life, cognitive impairment after treatment, or primary brain tumors were excluded. After a secondary analysis, reviewing the abstracts and/or full texts, 724 articles were found to be relevant. Publications significantly increased in the last 10 years. A total of 252 articles (34.8%) were published in 12 core journals, receiving 50% of the citations. The number of publications in Frontiers in Oncology, BMC Cancer, and Radiotherapy and Oncology have increased considerably over the last few years. There were 111 randomized controlled trials, 128 review articles, and 63 meta-analyses. Most randomized trials reported on brain metastases management from unselected tumors (49), lung cancer (47), or breast cancer (11). In the last 5 years (2017 to 2022), management of brain metastasis has moved on from WBRT, the use of chemotherapy, and radio-sensitization to three directions. First, Radiosurgery or Radiotherapy (SRS/SRT), or hippocampal-sparing WBRT is employed to reduce radiation toxicity. Second, it has moved to the use of novel agents, such as tyrosine kinase inhibitors (TKI) and immune checkpoint inhibitors (ICI) and third, to the use of molecularly directed therapy such as TKIs, in asymptomatic low volume metastasis, obviating the need for WBRT.

## 1. Introduction

The brain is a common site of metastases from cancer. Approximately 10% of all patients will develop brain metastasis during their illness [1]. Around 25% of patients with melanoma and lung cancer, and between 5 and 10% patients with breast cancer and renal cell carcinoma, may have brain metastases at the time of diagnosis. A significant number of patients develop brain metastasis within a year of their primary diagnosis [2,3]. Brain metastasis is frequently associated with significant morbidity and a shortened survival. 

Since the advent of CT scans, which enhanced the ability to detect and diagnose brain metastases, the treatment of brain metastases has evolved, and is evolving continuously. Whole brain radiotherapy (WBRT) used to be the mainstay of treatment [1,4,5]. However, adverse effects, such as late effects on cognitive function, emerged as a significant clinical challenge following WBRT. Furthermore, the prognosis for a significant number of patients with brain metastasis remained poor, with the median overall survival being 3–6 months. However, in a small proportion of patients, the survival was seen to extend for up to several years [6]. More recently, there have been attempts to balance improved survival with reducing the late side effects of radiotherapy, using stereotactic radiosurgery and stereotactic radiotherapy (SRS/SRT) techniques [7] and hippocampal sparing-WBRT [8,9]. In the last few years, various clinical factors, such as the performance status, the site of the primary tumor, the absence of the extra-cranial disease, and the biological features of the underlying cancer, have been shown to have prognostic significance [10,11,12]. For example, breast cancer patients have a variable clinical outcome if brain metastases develop in patients with hormone-receptor positive disease, HER-2/neu over-expressing disease, or triple-negative breast cancer [13,14,15,16]. Furthermore, the expression of cell surface markers and the molecular profile of the tumor allow tumors to be treated accordingly [17]. Therefore, the landscape of treatment of patients with brain metastases has changed and is changing rapidly.

Several reviews and meta-analyses have reviewed the efficacy of different modalities of treatment of brain metastases [18]. However, most meta-analyses and systematic reviews compared studies assessing the efficacy of novel treatments with the standard of care [19]. Similarly, systematic reviews also focus on the studies of outcomes of brain metastases either from a single tumor type [18,20,21] or different tumor types [18,22]. There are only a few reviews, who have comprehensively assessed the changing landscape of treatment, as reflected in the literature [23].

We conducted a systematic integrative review of the literature to inform the researchers and the clinical practitioners in the field of management of brain metastases. The objectives of this study were several-fold. First of all, to analyze the origin and pattern of the published literature on the subject, then to study the evolving landscape of management of brain metastases, and finally to identify current areas of active research in the field. 

## 2. Methodology

A systematic review of the published literature was carried out to identify the contents and patterns of the literature published on the management of brain metastases. Relevant articles on the management of brain metastases were searched for using the SCOPUS database. The search was performed on 2 September 2022 using the keywords “brain metastasis” OR “brain metastases” OR “CNS metastasis” OR “CNS metastases” AND “phase III” OR “Randomized Controlled Trial”. 

A total of 1986 documents were retrieved, published over a 45-year period (1977–2022) (Table 1). Ninety articles were excluded after restricting the subject area to medicine and nursing. Further, the search was restricted to ‘Journal Articles’, and another 261 articles were eliminated. Confining the search to ‘English language’ articles resulted in 1575 evaluable publications. At this stage, manual skimming was conducted to remove irrelevant articles. ‘Relevant articles’ were defined as clinical studies describing the treatment or prevention of brain metastases from any cancer. Published articles describing imaging modalities, quality of life, or cognitive impairment after treatment only, were excluded. Also, articles related to primary brain tumors were excluded. Manual skimming was performed by two authors, and wherever there was a discrepancy, the authors discussed the article to decide whether it was relevant or not.

A secondary analysis was carried out, by reviewing the abstracts or the entire text, resulting in the identification of articles not identified in the primary search. 

Eventually, to ensure related articles were included, all cited articles were compared with the reference list in the following studies: Mok, Wu [24], Murthy, Loi [25], Brown, Jaeckle [26], and Chang, Shi [27]. This process resulted in the identification of another 5 articles. Hence, the final sample size consisted of 720 articles published between 1977 and 2022, which form the basis of this analysis.

The data were plotted over time (number of publications or citations) and analyzed using Bradford’s law, Lotka’s law, and co-word analysis, using the “bibliometric package” developed in R-language version R 4.3.2 binary for macOS 11 (Big Sur) and higher, signed and notarized packages [28]. Pie-charts were used to show the distribution of categorical data, and cross tabulation was carried out to demonstrate the types of publications over the course of time (Figure 1).

## 3. Results

Over a period of 45 years (1977–2022), a total of 724 articles were identified as relevant for analysis. Figure 2 shows the number of papers published annually in the study period. The number of articles increased significantly in the last 10 years. The data from 2022 are not complete as the literature review was performed on 2 September 2022.

Journals which published the articles on the subject were classified into three zones, as described by Bradford’s law. This is shown in Figure 3A. A total of 252 articles (34.8%) were published in 12 core journals, receiving 50% of the citations. The details of the number of publications, the total citations received by the core journals, and their h-index are shown in Table 1.

Amongst the core journals, publication trends over the course of time are shown in Figure 3B. While increasing numbers of papers have been published in Frontiers in Oncology, BMC Cancer, Radiotherapy and Oncology, and Cochrane Database, the number of publications in other core journals has either remained stable or shown a downward trend in the last 10–15 years, suggesting a shift of focus of interest of those journals in the space of brain metastasis, or the pattern of submission to other journals. It is noteworthy that journals, such as BMC Cancer, The Lancet Oncology, and Frontiers in Oncology started publishing on the subject only in 2007, 2009, and 2012, respectively, however, they are ranked 7th, 5th, and 3rd in terms of number of publications, as shown in Table 1.

**Table 1 cancers-15-05570-t001:** Number of publications, citations, h-index, and the start year of the 12 core journals.

Journal	Number of Publications	Total Citations	h-Index	Publication Start Year
Journal of Clinical Oncology	37	8802	33	2000
International Journal of Radiation Oncology Biology Physics	32	5781	28	1997
Frontiers in Oncology	23	152	9	2012
Annals of Oncology	19	1474	15	1998
The Lancet Oncology	17	6336	16	2009
Journal of Neuro-Oncology	17	1110	15	1984
BMC Cancer	17	521	11	2007
Cancer	14	1515	13	1978
Radiotherapy and Oncology	14	470	12	1997
Radiation Oncology	13	348	10	2006
Clinical Oncology	12	418	9	1996
Journal of Neurosurgery	12	518	9	2006

The most prolific authors, the time when they started to publish on the management of brain metastasis and the number of publications every year are shown in Figure 4. Of the 10 most prolific authors, 2 authors (Mehta and Gaspar) have published on the subject over a period spanning more than 2 decades. 

The productivity of authors was studied using Lotka’s law [29], and is shown in Figure 4B. It appears that the pattern of the number of publications tends to follow Lotka’s law.

Next, we looked at the keyword clusters, as illustrated in Figure 5. The co-word analysis is a display of arrays of terms or combinations of words utilized in the published papers. It is used to analyze and visualize the relationship and interaction between topics. This analysis revealed several clusters; the major cluster (shown in the orange color) covers most of the publications. A few smaller clusters were related to primary brain tumors (excluded from this analysis), the use of trastuzumab in Her-2 over expressing breast cancer, ‘CNS’ metastasis in breast cancer, referring to a combination of brain parenchymal metastasis and leptomeningeal disease, the use of prophylactic cranial irradiation in small cell lung cancer, and the management of oligometastases. 

We then looked at the type of publications, and the results are shown in Figure 6. A total of 111 randomized clinical trials (RCTs) were identified over the period. The vast majority were phase III clinical trials; however, a few randomized phase II trials were also included. The number of systematic and narrative review articles and the number of meta-analyses is also shown in the figure. A large number of articles other than those three categories included publications describing single-center experiences, phase I and non-randomized phase II trials, cohort studies, case–control studies, radiation techniques in special circumstances, such as in the elderly population, hippocampal avoidance, the use of radiomics, proposal for future clinical studies, supportive care, cost-effectiveness, assessment of toxicity, neuro-cognitive side-effects and outcomes, quality management, management of cerebral edema, editorials, opinion papers, commentaries, clinical practice guidelines, etc. 

Of the 111 RCTs, 49 dealt with unselected solid tumors, 36 with non-small cell lung cancer (NSCLC), 11 with small cell lung cancer (SCLC), 11 with breast cancer, and 6 with melanoma. We then focused on the investigational arm of the RCTs to study the modality of treatment used in the investigational arm of the RCTs. As is clear from Figure 6B, the use of tyrosine kinase inhibitors (TKI), prophylactic cranial irradiation (PCI), stereotactic radiosurgery (SRS) with or without whole brain radiotherapy (WBRT), chemotherapy (CTX), and radio sensitization were the major interventions. A few RCTs also studied the addition of simultaneous integrated boost (SIB) or surgery to WBRT compared to WBRT alone. Finally, we studied the temporal pattern of publications in the management of brain metastases. Up until 2001, PCI, WBRT, and radio sensitization dominated as the subjects of RCTs. At the turn of the century, SRS began to be introduced, initially in combination with WBRT, and later on as the sole modality of treatment. In addition to SRS, systemic chemotherapy, and more recently TKIs have emerged as the preferred subjects of study in RCTs (Figure 7). 

## 4. Discussion

The aims of this study were to review the patterns of the published literature about the management of brain metastases arising from solid tumors, and to study the temporal pattern of evolution of the management of brain metastases over the last several years, as reflected in the publications. The study also analyzed the conceptual and social structure of the management of brain metastasis using a network analysis. A sharp increase in the number of publications was observed from 2012 onwards, and almost 50 papers were published annually on the subject over the last 10 years. The publication trend was consistent with Bradford’s law and Lotka’s law. A total of 111 RCTs were published in the time period. Of the RCTs, the vast majority dealt with brain metastasis arising from unselected solid tumors, followed by NSCLC, SCLC, breast cancer, and melanoma. In the last two decades of the last century, randomized trials establishing the role of WBRT dominated the landscape, and later, the management space was occupied by the addition of SRS, SIB, or radio sensitization to WBRT, followed by SRS alone, and more recently, the use of systemic chemotherapy or TKIs emerged as the predominant method of treatment of brain metastasis.

To review the evolution of the management of brain metastasis, we used an integrative review, which is a combination of a citation-based review and content analysis. The citation-based systematic literature review helped to study the pattern of publications, identification of core authors, journals, and keywords, whereas the content analysis provided categorical information from open-ended data, in terms of the type of cancer being studied and the type of investigational treatment, etc. An integrative review summarizes the literature to provide a more comprehensive understanding of a particular phenomenon or healthcare problem [30]. On the one hand, an integrative review allows for inclusion of diverse methodologies, review evidence, and gaps in the literature, but on the other hand, the combination and complexity of the subject can contribute to a lack of rigor [31]. Given the nature of the question in this study, i.e., temporal patterns of management of brain metastasis from solid cancers, this review provides an insight based on randomized trials. 

We reviewed the literature using Bradford’s law, which states that “If journals were arranged in order of decreasing productivity of articles on a given subject, they may be divided into a nucleus, particularly devoted to the subject, and several other zones containing the same number of articles as the nucleus” [32]. Bradford claimed that for a given subject, there are a few very prolific journals, a larger number of more moderate producers, and an even larger number of ever-decreasingly productive journals. For each issue or subject, the top third (zone 1 or core) journals are the most frequently cited in the literature and are likely to be of interest to researchers in the discipline. The middle third (zone 2) includes the journals with an average citation frequency, and the lower third (zone 3 or tail) includes a long tail of journals considered of marginal importance to the discipline and are rarely cited. Applying Bradford’s law to our data revealed that 12 journals belonged to the core zone, and three of those (Frontiers in oncology, BMC Cancer and Radiotherapy and Oncology) continued to publish an increasing number of articles over the last few years. 

We also reviewed the data using Lotka’s law. Lotka described the productivity of authors in the development of a particular field [29]. According to Lotka’s law, the relative frequency distribution of author productivity is predicted to be a hyperbolic inverse square function. It means that a small number of authors in a field publish most articles [33]. Also, authors publishing ‘n’ number of articles are approximately 1 n⁄2 of those publishing one article, and the number of authors who publish once only is about 60% (20). Our results were consistent with Lotka’s law.

The contents of 111 RCTs (Phase II and Phase III) were studied in more detail (73–187). Almost 85% of the RCTs compared the efficacy of one modality over another in patients with lung cancer, or unselected solid tumors, the majority of which were patients with lung cancer. This is not surprising, as the number of patients with lung cancer who develop brain metastasis is very high. Around 25–40% of patients with NSCLC develop brain metastases during their illness [34,35], more than 10% of patients with SCLC have brain metastases at the time of diagnosis [36], and several others develop brain metastasis during the course of their illness. The other two cancers in which brain metastases are common are breast cancer and malignant melanoma. 

The pattern of publications of RCTs reflects the evolution in the management of brain metastases. The last two decades of the last century were dominated by WBRT for treatment and PCI for prevention of brain metastases. WBRT has been the standard of care for treatment of brain metastases since the 1970s. However, the major concern was the potential for developing neurocognitive failure, the risk of which increases with longer patient survival due to better systemic treatment. At the turn of the century, SRS was introduced, and the first randomized trial was published in 2002. Ever since, SRS has been tested either together with WBRT (10 trials) or in comparison to WBRT (13 trials, most in the period from 2017 to 2022). The largest trial in the first category, WBRT with or without SRS, revealed improved local control via the addition of SRS boost, but no significant increase in survival (6.5 months vs. 5.7 months). Patients with 1–3 newly diagnosed brain metastases were treated by either WBRT or WBRT followed by SRS boost. However, patients with Disease Specific-Grade Point Assessment (DS-GPA) 3.5–4.0 had a better OS when treated with WBRT + SRS [6]. The median survival time was 21.0 months compared to 10.3 months for WBRT alone. An SRS boost resulted in improved survival in a subset of patients with a single brain metastasis (21.0 vs. 11.4 months) in the GPA 3.5–4.0 group. The use of SRS/SRT alone, without WBRT, is associated with a better preservation of neurocognitive function. Hence, in a different study design, WBRT and SRS were compared to SRS/SRT alone. The use of WBRT plus SRS did not improve survival for patients with one to four brain metastases; however, intracranial relapse occurred more frequently in those who did not receive WBRT. Consequently, salvage treatment was required frequently when up-front WBRT was not used. Taken together, these results have led to a change in the practice in several centers. SRS/SRT is preferred in patients when eligible, depending on the size, number, and total volume of metastasis. In the case of an intracranial relapse, SRS/SRT may be used again if required, reserving WBRT for those who are not eligible for SRS/SRT.

Radio-sensitization using either nitrogen mustard compounds [37], metronidazole [38], motexafin gadolinium [39,40,41], efaproxiral [42,43], or chloroquine [44] was tested, without much success.

In the last 10 years, five major themes emerged. Two forms of radiotherapy were used, the use of WBRT with hippocampal avoidance [45,46] and SRS [47,48,49,50,51,52], especially when the number of brain metastases were either less than four, or more recently even with a higher number of brain metastases. SRS was used either as standalone modality or in combination with resection. Other than radiotherapy, three forms of systemic treatment also emerged in the last 10 years. Although traditional cytotoxic chemotherapeutic agents either do not cross the blood–brain barrier, or the metastasis are resistant to these agents, a total of 15 randomized trials reported the efficacy of using either chemotherapy as a single agent or in combination with WBRT. Temozolomide, although active in the management of glioblastma multiforme, was not found to be effective in the management of brain metastasis [53,54,55,56]. More recently, three trials were reported on the use of etrinotecan pegol [57,58,59]. Three trials compared the efficacy of immune checkpoint inhibitors in patients with melanoma or NSCLC [60,61,62]. A higher number of patients achieved an intracranial response with a combination of ipilumomab and nivolumab in patients with melanoma with asymptomatic brain metastases. Furthermore, compared to fotemustine, ipilumomab and nivolumab improved long-term overall survival. 

The most significant change in the pattern of management has been the use of TKIs to prevent the onset of brain metastases in patients with NSCLC with activating mutations in either the epidermal growth factor receptor (EGFR) gene, in case of anaplastic lymphoma kinase (ALK) gene rearrangement, or in Her-2/neu overexpressing breast cancer. Alternatively, the TKIs were used to treat asymptomatic brain metastases arising from tumors with these mutations. Most trials reported on the use of first, second, or third generation TKI of ALK gene rearrangement (crizotinib, Ensartinib, Loralatinib, or Brigatinib), or increasing doses of the third generation TKI, Brigatinib [63,64,65,66,67]. Together, these trials revealed a response rate of 64–82% with Ensartinib, lorlatinib, and brigatinib compared to 21–23% with the first generation TKI crizotinib. The rate of progression reduced from around 18–24% down to 1–4% in 1 year, and there was a 2–3-fold increase in PFS with the use of new generation TKIs. Increasing the dose of brigatinib from 90 to 180 mg per day in crizotinib-refractory ALK-positive NSCLC resulted in a higher response rate and duration of response.

At least three trials reported on the use of TKIs in patients with Her-2 positive breast cancer. The HER2CLIMB trial compared trastuzumab with tucatinib or placebo in patients with locally advanced or metastatic breast cancer for response rates, reduction in the risk of intracranial progression, and death because of brain metastases [68]. Although clinical trials usually exclude patients with brain metastases, almost 50% of patients accrued in the trial had brain metastases. The combination of tucatinib and trastuzumab doubled the intracranial objective response rate, reduced the risk of intracranial progression or death by two-thirds, and reduced the risk of death by nearly one in all patients with brain metastases. The estimated 1-year intracranial PFS (CNS-PFS) was 40% in the tucatinib arm and 0% in the control arm. The estimated 1-year CNS-PFS was 35% in the tucatinib arm and 0% in the control arm in the untreated patients, 53% vs. 0%, respectively, in patients with treated and progressing brain metastases [68]. An earlier trial comparing lapatinib and capecitabine vs. trastuzumab and capecitabine was terminated early because of poor recruitment and a similar response rate and PFS in the 40 patients randomized until the time of analysis [69]. A later trial on the use of pyrotnib and capecitabine revealed a response rate of 74% in patients known to have brain metastases from Her-2 positive breast cancer. A total of 16 such RCTs were identified, 12 were in the last 4 years [25,63,64,65,66,67,68,69,70,71]. 

Patients with brain metastasis are a heterogeneous group, with a short median survival. Furthermore, the occurrence of brain metastases is usually accompanied by severe morbidity and a substantial deterioration in quality of life. WBRT, SRS, or surgical resection are used for palliation, but may cause further deterioration in the quality of life. The introduction of TKIs has made it possible to individualize the therapy based on the presence of activating mutations in the EGFR gene, ALK rearrangement, or tumors harboring alterations in RET and MET genes. 

There are several limitations of this analysis. Firstly, only SCOPUS was searched to identify published articles. SCOPUS is commonly used to carry out citation-based systematic literature reviews (Sainaghi, Köseoglu [72], Ahmad, Naveed [73], Geetha and Kothainayaki [74,75,76]), as it includes the widest range of articles with complete reference sets in a consistent and reliable form [77]. Another limitation is that all publications related to the management of brain metastasis may not have been identified; however, we identified more than 1000 articles related to using the specific keywords and then limited the search to 720 articles for the final analysis. Yet another limitation is publication bias, which is inherent in all review articles. The purpose of our study was to review the published studies only to assess the evolving patterns of care of brain metastases. 

## 5. Conclusions

In conclusion, this integrative review shows an exponential increase in the number of publications on the subject of management of brain metastases, especially those arising from lung and breast cancers and melanoma. RCTs suggest that the emphasis of management is moving into three major directions, reducing the toxicity of WBRT either by using SRS/SRT or hippocampal sparing WBRT, the use of novel agents such as, ICI and newer forms of chemotherapy, and the use of TKIs for prevention as well as the treatment of asymptomatic small volume brain metastases. 

## Figures and Tables

**Figure 1 cancers-15-05570-f001:**
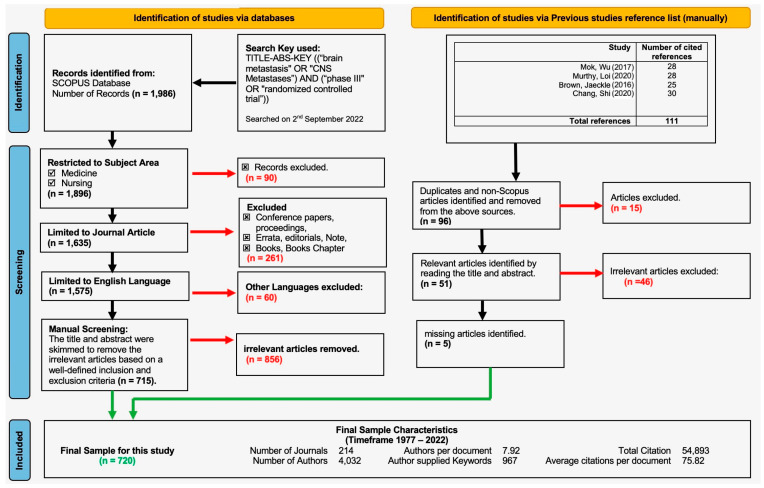
Article selection flow chart (PRISMA) [24,25,26,27].

**Figure 2 cancers-15-05570-f002:**
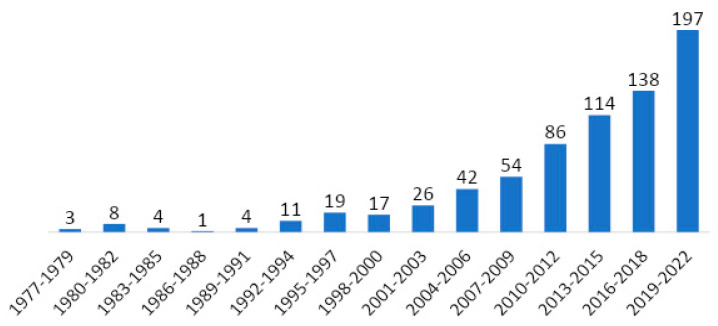
Publication trends over the course of time.

**Figure 3 cancers-15-05570-f003:**
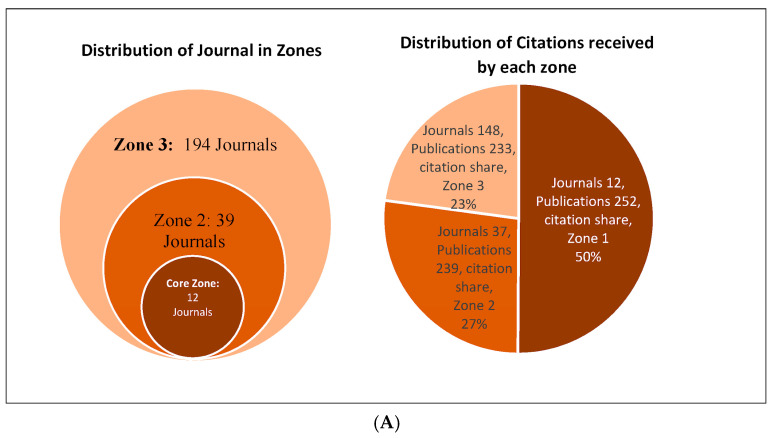
(**A**) Bradford’s law, Core journals publish almost 1/3rd of all articles, and the journals in zone 2 and zone 3 also publish 1/3rd of all articles each. However, articles published in the 12 core journals receive almost 50% of all citations. (**B**) Publication trend over the course of time in 12 core journals.

**Figure 4 cancers-15-05570-f004:**
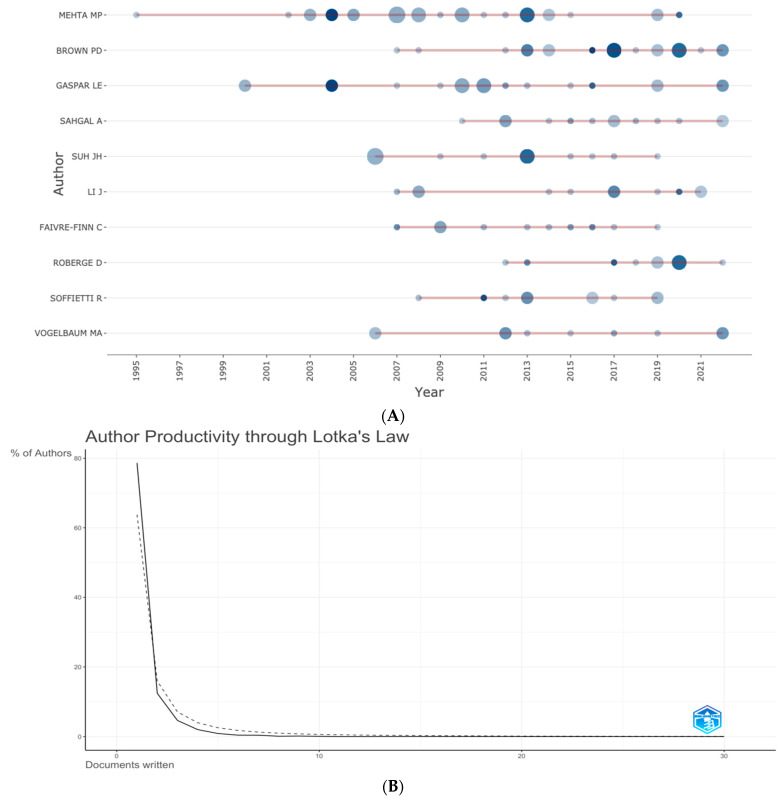
(**A**) Author production over time. Red line indicates the author′s timeline, the bubble size indicates the number of documents published that year, and the color intensity indicates the total number of citations per year. (**B**) Author productivity using Lotka’s law. Solid black line indicates the distribution of published articles according to Lotkaw’s law. The dotted line indicates the publication on subject matter. The solid and the dotted lines almost overlap, suggesting the publication trend in management of metastases from breast cancer follows the law.

**Figure 5 cancers-15-05570-f005:**
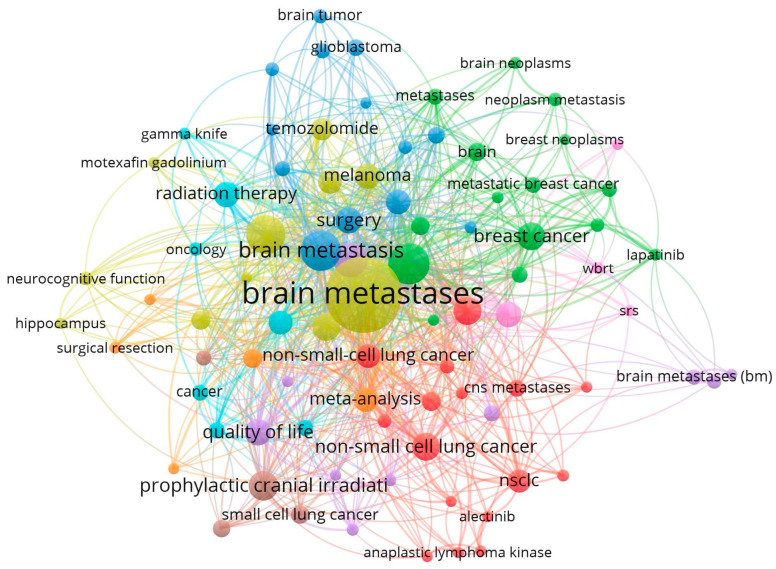
Author’s keyword clusters.

**Figure 6 cancers-15-05570-f006:**
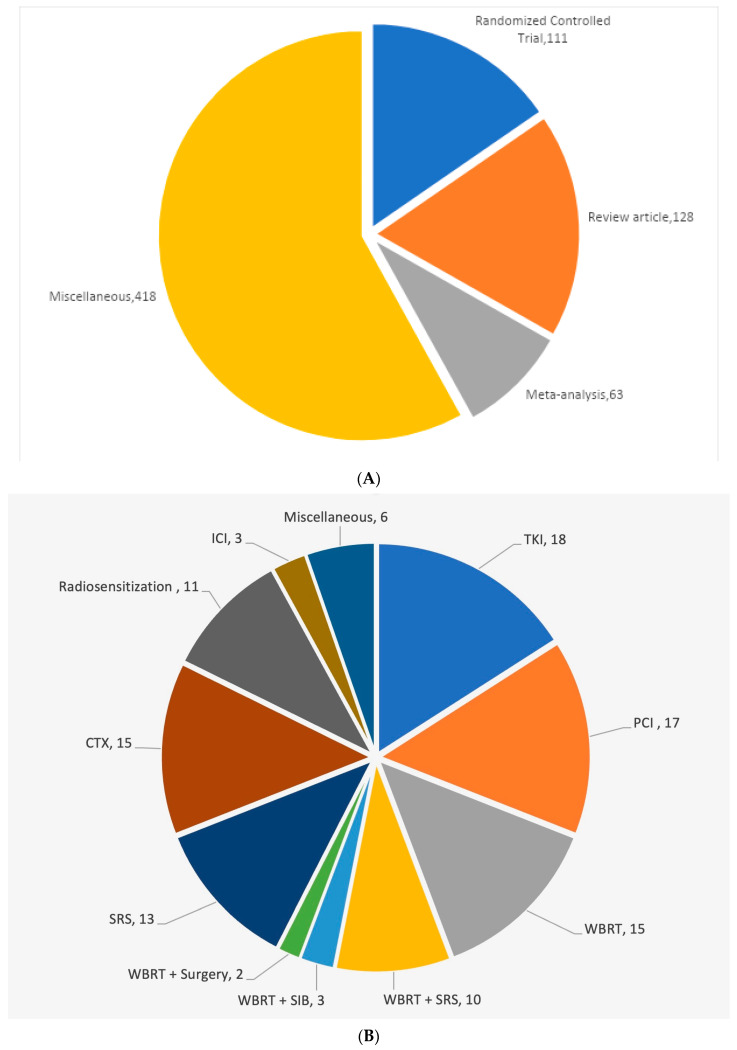
(**A**) Nature of publications. (**B**) Investigational arm of the randomized control trials.

**Figure 7 cancers-15-05570-f007:**
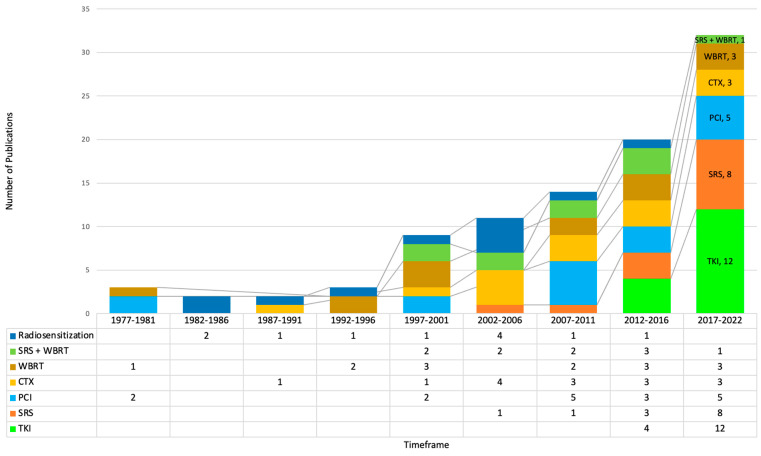
Temporal pattern of the modalities of treatment introduced for the management of brain metastases.

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
