# Peer review of "Evolution of the Management of Brain Metastases: A Bibliometric Analysis"

_cancers, 2023, doi:10.3390/cancers15235570_

Round 1

Reviewer 1 Report

Comments and Suggestions for Authors

The authors performed literature search to identify trials that investigated brain metastasis treatment and concluded the evolutionary change and increasing complex treatment patterns for brain metastasis.  The authors should note that inherent publication bias is present.  Another limitation is that the authors did not attempt to analyze the effectiveness of treatment, but instead focused on reporting treatments being investigated in different period of time.  It is not clear how this derive an objective information that may be benefit practitioners.  IT is suggested that the authors examine the results of the studies and make inference and synthesize a finding from this analysis to provide clinical context of the analysis.

Author Response

Dear reviewer,

Please see below point-by-point response to the valuable comments. The comments were very helpful, and certainly helped to enhance the quality of article. The vast majority of comments have been incorporated into the updated version. The additional text is marked in red color. We also reviewed the entire manuscript for English language and made changes, shown as track changes.

Best wishes

Reviewer comment and authors reply:

The authors performed literature search to identify trials that investigated brain metastasis treatment and concluded the evolutionary change and increasing complex treatment patterns for brain metastasis. 

*Authors Reply*: Thank you for this kind comment.

The authors should note that inherent publication bias is present. 

*Authors Reply*: Thank you for this comment. The section on limitation of the study has been expanded and the comment has been included.

Another limitation is that the authors did not attempt to analyze the effectiveness of treatment, but instead focused on reporting treatments being investigated in different period of time.  It is not clear how this derive an objective information that may be benefit practitioners.  IT is suggested that the authors examine the results of the studies and make inference and synthesize a finding from this analysis to provide clinical context of the analysis.

*Authors Reply*: Thank you for this comment. Although the primary objectives of the bibliometric analysis were to report the type of published literature, the origin of publications, the pattern of evolution of treatment reflected in the literature, and to identify the directions for work in near future, however, in response to the comments of the learned reviewer, paragraphs have been added in the ‘discussion’ section, commenting on the effectiveness of various types of common interventions.

Reviewer 2 Report

Comments and Suggestions for Authors

I think this review is sufficiently complete and suitable for publication. 

Author Response

Dear reviewer,

Thank you so much for your comment.

Best wishes

Reviewer comment and authors reply:

I think this review is sufficiently complete and suitable for publication. 

*Authors Reply*: Thank you for your very kind comment

Reviewer 3 Report

Comments and Suggestions for Authors

The authors have covered a wide number of publications which is a plus point. The figures and diagrams are nicely done. The language of the paper needs to be refined and scientific. If authors could have analyzed any branch of brain cancer management in detail like meta-analysis would be great. The major drawback of the paper is that it lacks the personal opinions of the authors and a flow in writing is missing. Also, the manuscript only puts more attention to the data of previous publications rather than their essence in the management of disease. This makes the article less interesting to the readers.

Author Response

Dear reviewer,
Please see below point-by-point response to the valuable comments. The comments were very helpful, and certainly helped to enhance the quality of article. The vast majority of comments have been incorporated into the updated version. The additional text is marked in red color. We also reviewed the entire manuscript for English language and made changes, shown as track changes.
Best wishes
On behalf of the authors

Reviewer comment and authors reply:

The authors have covered a wide number of publications which is a plus point. The figures and diagrams are nicely done.

*Authors Reply*: Thank you for your comment

The language of the paper needs to be refined and scientific.

*Authors Reply*: Thank you for your comment. The entire manuscript has been proof-read and English language editing has been done

If authors could have analyzed any branch of brain cancer management in detail like meta-analysis would be great.

*Authors Reply*: Thank you for this comment. Although the primary objectives of the bibliometric analysis were to report the type of literature, the pattern of evolution of treatment reflected in the literature, and to identify the directions for work in near future, however, in response to the comments of the learned reviewer, paragraphs have been added in the ‘discussion’ section, commenting on the effectiveness of various types of common interventions. A meta-analysis would be out of the scope of this paper, as the literature covers a period of 45 years, and the treatment modalities span from primary prevention (PCI), to secondary prevention (TKIs), to treatment of asymptomatic brain metastases (TKIs), to treatment of symptomatic brain metastases (WBRT, and SRS, etc)

The major drawback of the paper is that it lacks the personal opinions of the authors and a flow in writing is missing.

*Authors Reply*: Thank you for your comment. The entire manuscript has been proof-read and English language editing has been done. Hopefully, the learned reviewer will find additional information in the discussion section and a better flow of writing.

Also, the manuscript only puts more attention to the data of previous publications rather than their essence in the management of disease. This makes the article less interesting to the readers.

*Authors Reply*: Thank you for this comment. Although the primary objectives of the bibliometric analysis were to report the type of published literature, the origin of publications, the pattern of evolution of treatment reflected in the literature, and to identify the directions for work in near future, however, in response to the comments of the learned reviewer, paragraphs have been added in the ‘discussion’ section, commenting on the effectiveness of various types of common interventions

Reviewer 4 Report

Comments and Suggestions for Authors

The title of the work: Management of Brain Metastases: An Integrative Review.

It is a very special summary about the care of brain metastases (BM), collecting all relevant publications is this topic, however the final message is not about the „best treatment options”, but it shows rather a longitudinal trend in the analyzed treatment categories. This kind of summary does not present a real clinical novelty/recommendation, nevertheless this a very special and interesting consideration of this important topic. Figure 7. masterfully illustrates this kind of consideration.

The material is a well written, excellent work, the reviewer has only a few questions/ concerns/ suggestions:

1, Title: If it is possible, I recommend a longer title with the type of analysis.

2, Abstract:

-The authors listed 3 core journals (Frontiers in Oncology, BMC Cancer, and Radiotherapy and Oncology), but it is not mentioned that these are the best examples for medical papers with increasing number of articles dealing with BM.  

-Both in the Abstract and in the Methodology the authors emphasized that „articles on imaging, quality of life, 37 cognitive impairments after treatment, or primary brain tumors were excluded”. However, some articles were dealing with these categories incidentally as well (see page 10). Please clarify this provincial issue.

-In the last sentence of the abstract the authors listed the most relevant novel issues in the management of BM, however I missed some words about the main topics of the past times (like the role of WBRT, radio-sensitization etc.).

3, Introduction:

The authors stated (lines 53-55.): „Most common primary cancer for brain metastasis at the time of diagnosis is Lung Cancer (25%), Melanoma (25%), Breast Cancer (10%) and Renal Cell Carcinoma (10%). I think it is true in case of primary metastatic cancer cases. Please clarify it.

And (lines 55-56.): „A significant number 55 of patients develop brain metastasis within a year of their primary diagnosis.” It is true only advanced cancer diseases. Please clarify it.

And (line 66.): „However, in a proportion of patients, the survival may extend up to several years.” It is true in case for oligometastatic cases and for good condition patients. Please amend it.

Finally, I recommend reducing the size of the figures to better survey the material.

Author Response

Dear reviewer, 
Please see below point-by-point response to the valuable comments. The comments were very helpful, and certainly helped to enhance the quality of article. The vast majority of comments have been incorporated into the updated version. The additional text is marked in red color. We also reviewed the entire manuscript for English language and made changes, shown as track changes.
Best wishes
On behalf of the authors

Reviewer comment and authors reply:

It is a very special summary about the care of brain metastases (BM), collecting all relevant publications is this topic, however the final message is not about the „best treatment options”, but it shows rather a longitudinal trend in the analyzed treatment categories. This kind of summary does not present a real clinical novelty/recommendation, nevertheless this a very special and interesting consideration of this important topic. Figure 7. masterfully illustrates this kind of consideration.

*Authors Reply*: Thank you for your comment

The material is a well written, excellent work, the reviewer has only a few questions/ concerns/ suggestions:

*Authors Reply*: Thank you for your comment

1, Title: If it is possible, I recommend a longer title with the type of analysis.

*Authors Reply*: Thank you for your comment. Title has been modified

2, Abstract:

-The authors listed 3 core journals (Frontiers in Oncology, BMC Cancer, and Radiotherapy and Oncology), but it is not mentioned that these are the best examples for medical papers with increasing number of articles dealing with BM.  

*Authors Reply*: Thank you for your comment. Please allow us to clarify this point. Figure 3A in the text shows the 12 core journals, which Published almost 1/3rd of all papers. Figure 3B shows the pattern of growth over the course of time. The three journals mentioned in the abstract are among the core journals, publishing more papers in the field over the last few years. The text in abstract section has been modified.  

-Both in the Abstract and in the Methodology the authors emphasized that „articles on imaging, quality of life, 37 cognitive impairments after treatment, or primary brain tumors were excluded”. However, some articles were dealing with these categories incidentally as well (see page 10). Please clarify this provincial issue.

*Authors Reply*: Thank you for your comment. Please allow us to clarify this point. A ‘relevant publication was defined as a published paper which deals with management of metastatic brain cancer. Some studies / papers had more than one end-point or angle to review including, for example, quality of life issues, effect of cognitive function as well.

-In the last sentence of the abstract the authors listed the most relevant novel issues in the management of BM, however I missed some words about the main topics of the past times (like the role of WBRT, radio-sensitization etc.).

*Authors Reply*: Thank you for your comment. The sentence has been amended.

3, Introduction:

The authors stated (lines 53-55.): „Most common primary cancer for brain metastasis at the time of diagnosis is Lung Cancer (25%), Melanoma (25%), Breast Cancer (10%) and Renal Cell Carcinoma (10%). I think it is true in case of primary metastatic cancer cases. Please clarify it.

*Authors Reply*: Thank you for your comment. The sentence has been amended.

And (lines 55-56.): „A significant number 55 of patients develop brain metastasis within a year of their primary diagnosis.” It is true only advanced cancer diseases. Please clarify it.

*Authors Reply*: Thank you for your comment. The sentence has been amended.

And (line 66.): „However, in a proportion of patients, the survival may extend up to several years.” It is true in case for oligometastatic cases and for good condition patients. Please amend it.

 *Authors Reply*: Thank you for your comment. The sentence has been amended.

Finally, I recommend reducing the size of the figures to better survey the material.

*Authors Reply*: Thank you for your comment. In our opinion, the type-setting process will take care of this.

Round 2

Reviewer 1 Report

Comments and Suggestions for Authors

The authors expanded the discussion session which adds more substance.  Nevertheless, the major weakness is the design of the study, that the authors did not synthesize results but merely collected studies published over time and focused on the impact of the journals, not the scientific merit.